# Monitoring Therapy Efficiency in Cancer through Extracellular Vesicles

**DOI:** 10.3390/cells9010130

**Published:** 2020-01-06

**Authors:** Ines Stevic, Gustav Buescher, Franz Lennard Ricklefs

**Affiliations:** 1Department of Neurosurgery, University Medical Center Hamburg-Eppendorf, 20251 Hamburg, Germany; i.stevic@uke.de; 2I. Department of Medicine, University Medical Centre Hamburg-Eppendorf, 20251 Hamburg, Germany; g.buescher@uke.de

**Keywords:** extracellular vesicles (EVs), cancer, therapy monitoring, chemoresistance

## Abstract

Extracellular vesicles (EVs) are a heterogeneous group of membrane-enclosed vesicles made of a phospholipid bilayer and are secreted by all cell types. EVs are present in a variety of body fluids containing proteins, DNA, RNA species, and lipids, and play an important role in cell- to-cell communication and are worth being considered as biomarkers for both early diagnosis of cancer patients and real-time monitoring of treatment response. Recently, emerging evidence verified EVs to have crucial roles in cancer progression and metastasis and a great potential in therapeutic applications. In this review, we discuss the potential of EVs in monitoring the efficacy of cancer therapies.

## 1. Background

Extracellular vesicles (EVs) are a heterogeneous population of lipid bilayer membrane vesicles that are released by almost all cell types. Various subpopulations of EVs such as exosomes, microvesicles, microparticles, ectosomes, oncosomes, and many others have been well described. Based on their origin, it is most likely that they carry distinct biological functions [1]. Due to their ability to protect and transfer biological cargo consisting of proteins, lipids, and nucleic acids to recipient cells, EVs currently emerge as a crucial player in horizontal cellular communication [2,3,4]. The EV shuttle, as a process of communication, can modify target cells and their functions at close or distant range. EVs are present in every human body fluid including blood, urine, amniotic fluid, breast milk, seminal fluid, saliva, lymphatic fluids, cerebrospinal fluid, and bile, generating an optimal source for liquid biopsy approaches [5]. An increasing amount of research reports that EVs are potential biomarkers in different types of cancer [6]. The current state of a cell during cancer development and treatment is reflected by actively secreted EVs, providing valuable real-time information on dynamic changes [7,8].

Liquid biopsy is a non-invasive way of obtaining cancer-derived components that may enable real-time patient monitoring and the response to treatment. These cancer-derived components include circulating tumor cells (CTC), circulating cell-free tumor DNA, microRNA, long non-coding RNA, and EVs [9]. MicroRNAs (miRNAs) are a class of small non-coding RNA molecules with approximately 18–25 nucleotides in length. They regulate gene expression post-transcriptionally by binding to the 3′untranslated region (UTR), coding sequences, or 5’UTR of target mRNAs, leading to inhibition of translation or mRNA degradation [10,11]. MiRNAs may have dual functions and can be both tumor suppressors and oncogenes. The binding of miRNAs to target mRNA is essential for regulating mRNA level and protein expression. Cancer-derived EVs are enriched in miRNA of their specific origin and contain rare yet highly specific RNA biomarkers. Being one of the major components of EV cargo, investigation of RNA species and their potential as liquid biopsy is evident and has been achieved by numerous studies [12,13,14,15]. Besides RNA species, DNA was also discovered to be present in EVs, containing the entire genome and with it genetic mutations of the parental tumor cell with the potential for real-time analysis of the cancer’s genetic status during therapy.

In addition, proteome analysis of isolated tumor EVs may significantly reduce noncancer specific proteins, thereby increasing signal-to-noise ratio for biomarker identification.

In conclusion, the analysis of circulating EV cargo is a potential tool to monitor tumor cell changes during anti-tumor therapy (summarized in Figure 1), which we will discuss in this review.

## 2. EVs and Immune Checkpoint

Immune checkpoints play an important role in immune regulation and blocking these checkpoints on tumor cells is nowadays widely used in novel anti-cancer treatment regimens [16]. The necessity to monitor immunotherapy has increased the interest in the potential of EVs as a regulating tool for checkpoint therapy. Recent studies have shown that the well-known programmed cell death 1 ligand (PD-L1) is present on EVs and can systemically suppress anti-tumor immunity. Ricklefs et al. [17] (Table 1) showed that glioblastoma (GBM) EVs can block T cell activation and proliferation in response to T cell receptor stimulation. GBM EVs that were PD-L1^high^ significantly inhibited the magnitude of T cell activation and this was partially reversed by PD1 blockade. Additionally, PD-L1 DNA from circulating EVs in patients correlated with GBM tumor volume. Accordingly, EVs PD-L1 might be one mechanism of GBM to suppress antitumor immunity, underlining the potential of EVs as biomarkers in tumor patients [17].

A later study by Chen et al. [18] (Table 1) analyzed PDL1-associated EVs (exosomes) in melanoma. The experiment was primarily accomplished in vivo using a syngeneic mouse melanoma model in C57BL/6 mice and B16-F10 cells in which PD-L1 expression was present or had been knocked down. After EV injection in mice, they observed a significant decrease in the number of tumor-infiltrating CD8+ T-lymphocytes (TILs) in the PD-L1 expressing group. Their analysis of patients with metastatic melanoma showed a positive correlation of EVs PD-L1 levels and interferon-γ (IFN-γ) and varied during anti-PD-L1 therapy. The level of EVs PD-L1 before treatment was significantly higher in patients who did not respond to anti-PD-L1 treatment with pembrolizumab, associated with poor clinical outcome. Their study indicated that EVs could potentially be used for monitoring of PD-L1 therapy in melanoma patients [18].

A study from Theodoraki et al. [19] (Table 1) analyzed whether tumor-derived exosomes (TEX) and/or T-cell-derived EVs could predict outcome in head and neck squamous cell carcinoma (HNSCC) patients treated with ipilimumab. Patients pre- and post-therapy were included. In patients with recurrent disease, TEX/total EVs ratios, total CD3+, CD3(−)PD-L1+, and CD3+ CD15s+ (Treg-derived) exosomes increased from baseline levels. In contrast, in patients who remained disease free, total EVs protein and TEX levels decreased, CD3+ and CD3+ CD15s+ exosomes stabilized, and CD3+ CTLA4+ exosomes declined after ipilimumab therapy. In summary, they showed that TEX and T-cell-derived circulating exosomes can be used to monitor response to oncological therapy [19].

## 3. EVs in Breast Cancer Therapy Monitoring

Breast cancer (BC) remains the most frequent cancer among women [20]. The purpose of administering neoadjuvant chemotherapy (NACT) prior to surgery is to downstage the tumor allowing less extensive surgery. In addition, response to NACT is an early evaluation of the effectiveness of subsequent systemic therapy. Overall absence, presence, or even extent after NACT is a strong prognostic factor for risk of recurrence [21]. Patients with a pathologic complete response (pCR) after NACT have a significant higher overall and disease-free survival (OS, DFS) than patients with residual invasive disease [22]. Until now, both on-treatment and post-NACT evaluation were based on clinical examination, imaging (ultrasound, MRI), and histology. The correlation between clinical assessment and pathological analysis after NACT was modest at best. Several studies showed a low sensitivity to predict pCR by clinical examination and imaging [23]. Due to the lack of concordance, a need for reliable biomarkers that allow non-invasive therapy monitoring and stratification of patients became evident.

König et al. (Table 1) analyzed circulating EVs counts as additional markers for disease monitoring and prediction of prognosis in primary, non-metastatic, locally advanced breast cancer (BC) patients. EVs were analyzed together with circulating tumor cells (CTCs), and isolated from plasma samples of BC patients before and after NACT. EV concentration increased during therapy and an elevated EV concentration before NACT was associated with lymph node infiltration, and elevated EV concentration after NACT was associated with a reduced three-year progression-free and overall survival. Both EVs and CTCs from one sample comprised different but complementary information on BC disease status and prognosis. Thus, the authors propose to use EVs as an additional parameter in assessing minimal residual disease as well as therapy and disease outcome in parallel with CTC analysis [24].

Many studies confirm primary tumors to release EVs actively, thus helping the formation of pre-metastatic niche and enhancing metastasis [1,25,26,27]. Keklikoglou et al. [28] (Table 1) analyzed the effect of taxanes and anthracyclines, commonly used cytostatics in neoadjuvant treatment of BC, on secreting tumor-derived EVs with enhanced pro-metastatic capacity. They showed that chemotherapy-secreted EVs could promote NF-kb-dependent endothelial cell activation in an Annexin A6 dependent manner and induce CcL2 as well as and Ly6C^+^CCR2^+^ monocyte expansion, resulting in the establishment of lung metastasis of BC cancer cells. In the future, EVs from chemotherapy educated tumor cells might provide a biomarker predicting the risk of metastasis in patients who do not achieve a complete response with neoadjuvant chemotherapy [28].

Shen et al. [29] (Table 1) showed that the treatment with a sublethal dose of chemotherapeutic agents induced BC cells to secrete EVs and stimulate a cancer stem-like cell (CSC) phenotype, and cancer cells became resistant to therapy. It was shown that EV miRNAs, like miR-9-5p, miR-203a-3p, and miR-195-5p, regulate CSC-associated phenotype through directly targeting the transcription factor One Cut Homeo-box 2 (ONECUT2), subsequently causing induction of CSC traits and expression of stemness-associated genes, including NOTCH1, SOX9, NANOG, OCT4, and SOX2. ONECUT2 was proposed as a common target of the five miRNAs, and its lower expression was associated with worse relapse-free survival of BC patients. The authors showed a mechanism through which chemotherapy-treated breast cancer cells, by secreting certain EV miRNAs, communicate with and reprogram nearby cancer cells to induce a CSC phenotype. Since there are well-known associations between CSCs and tumor refractoriness, this mechanism may serve as a means for cancer’s self-adaptation to survive the therapy, and may contribute to chemotherapy-induced tumor progression and metastasis. Targeting these adaptation mechanisms along with chemotherapy represents a potential strategy to maximize the anticancer effect and to reduce chemoresistance in cancer management [29].

Chen et al. [30] (Table 1) analyzed the relationship between breast cancer resistance protein (BCRP) and circulating EVs (microvesicles). The group showed that the levels of BCRP in patients who did not respond or had progressive/stable disease following chemotherapy were higher compared to those that did not receive chemotherapy. BCRP was found to be upregulated at the mRNA and protein levels in circulating EVs from cancer patients that had a poor response to chemotherapy. They also found that BCRP and flotillin-2 were upregulated in tumor samples from nonresponsive patients [30]. Flotillin-2 has been shown to be involved in various cellular processes such as cell adhesion and signal transduction through receptor tyrosine kinases as well as in cellular trafficking pathways [31]. Flotillin-2 is a known EV marker. Thus, to focus more on association between BCRP and EVs, the BCRP levels were assessed in circulating EVs isolated from patients at the mRNA and protein levels. Currently tumor marker such as CA15-3, carcinoembryonic antigen, and erythrocyte sedimentation rate are usually used to determine clinical response to chemotherapy [32]. Nevertheless, these markers did not accurately estimate patient response and lacked sensitivity revealing the need for improved markers. Tumor-derived circulating EVs that carry BCRP might provide a predictive biomarker for the response to chemotherapy of breast cancer [30].

Van Dommelen et al. [33] (Table 1) reported that cetuximab treatment alters the protein content of EVs. EV levels of epidermal growth factor receptor (EGFR) and phospho-EGFR were reduced after cetuximab treatment, reflecting similar changes in the parental cells. EV-associated cetuximab reduced EGF-mediated activation of kinases in human umbilical vein endothelial cells. It was shown that therapy with cetuximab can be monitored via EVs [33].

**Table 1 cells-09-00130-t001:** Summary of studies investigating extracellular vesicles (EVs) in therapy monitoring.

Type of Cancer	Monitored Therapy	Bio-Fluid EV Isolation Technique	Read-Out/Method	Comment	Reference
Glioblastoma	Immune checkpoint therapy	Blood, cell culture supernatant UC	ELISA and ddPCR	Glioblastoma EVs blocked T cell activation and proliferation in response to T cell receptor stimulation. PD-L1 was expressed on the surface of glioblastoma-derived EVs with potential to directly bind to PD1	[17]
Melanoma	Immune checkpoint therapy	Blood, cell culture supernatant UC	FACS and ELISA	metastatic melanoma releases a high level of EVs that carry PD-L1 on their surface. Interferon-γ (IFN-γ) up-regulates PD-L1 on these vesicles, which suppresses the function of CD8 T cells and facilitates tumor growth.	[18]
Head and neck squamous cell carcinoma	Chemotherapy radiation	Blood, cell culture supernatant SEC	FACS and microarrays	Disease recurrence was associated with increase of total exosome proteins, after ipilimumab therapy, total exosome protein and tumor-derived and/or T-cell derived exosomes levels decreased, CD3+ and CD3+ CD15s+ exosomes stabilized and CD3+ CTLA4+ exosomes declined.	[19]
Breast cancer	Neo-adjuvant chemotherapy	Blood PEG based	Levels of EVs and CTCs were analyzed	Increased EVs concentration pre chemotherapy was associated with therapy failure and elevated EV concentration post-chemotherapy was associated with a reduced three-year progression-free and overall survival.	[24]
Breast cancer	Chemotherapy	Blood, cell culture supernatant UC	Protein levels and in vivo study	Chemotherapy-elicited EVs were enriched in annexin A6 (ANXA6).	[28]
Breast cancer	Chemotherapy	Cell culture supernatant, blood UC	miRNA	Chemotherapeutic agents induced breast cancer cells to secrete EV with the capacity to stimulate a cancer stem-like cell phenotype, promoting resistance to therapy	[29]
Breast cancer	Chemotherapy	Tissue and blood UC	mRNA and protein level	Breast cancer resistance protein was found to be upregulated at the mRNA and protein levels in circulating EVs from cancer patients that had a poor response to chemotherapy	[30]
Breast cancer	Immunotherapy	Cell culture supernatant UC	Protein level	Cetuximab treatment altered the protein content of EVs.	[33]
Glioblastoma	Surgical resection	Cell culture supernatant UC	mRNA and miRNA	serum (microvesicles) contain messenger RNA mutant/variants and microRNAs characteristic to patients with glioblastomas	[34]
Glioblastoma	Surgical resection	Blood UC	Mass spectrometry	EVs concentration was higher in GBM compared with healthy Controls, brain metastases and extra-axial brain tumors. Significant drop in plasma concentration was measured after surgery	[35]
Prostate cancer	Radiotherapy	Blood PEG based	miRNA PCR	higher vesicle concentration of exosomes and upregulation of hsa-let-7a-5p and hsa-miR-21-5p indicating radiation specific induction	[36]
Prostate cancer	Radiotherapy	Blood PEG based	miRNA next-generation sequencing	miR-654-3p and miR-379-5p expression after radiotherapy was associated with therapy response	[37]
Prostate cancer	Antiandrogen therapy	Cell culture supernatant UC	proteomics	vesicular protein cargo (ATP2B1/PMCA ATPase) possibly mediates resistance towards hormone therapy	[38]
Colorectal carcinoma	Surgical resection	Blood centrifugation of MPs	FACS from taMPs	EpCAM+ taMPs decreased 7 days after curative R0 tumour resection suggesting therapy succes	[39]
Liver cancer	Surgical resection	Blood centrifugation of MPs	FACS from taMPs	taMPs positive for AnnexinV, EpCAM and ASGPR1 decreased 7 days after curative R0 tumour resection	[40]
Colorectal carcinoma	Surgical resection	Blood UC	miRNA	low levels of vesicular miR-200c and miR-141 were associated with longer OS after CRC resection	[41]
Colorectal carcinoma	Radiotherapy	Blood CellSearch Imaging Flow Cytometry, Menarini^®^	relative changes in total number of CTCs, MPs, cell fragments	combination of relative changes in the total number of CTCs, MPs, and cell fragments together with perfusion CT scan classifies patients as responders or non-responders	[42]
Pancreatic cancer	Surgical resection	Blood UC	FACS	GPC1+ EVs significantly decreased after surgical resection, implacting therapy response	[43]
Pancreatic cancer	Neoadjuvant chemotherapy	Blood UC	exosomal DNA KRAS mutation ddPCR	vesicular KRAS mutation after neoadjuvant therapy is associated with disease progression and no option for surgical intervention; a reduction correlates with resectability	[44]
Lymphoma	Chemotherapy	Blood SEC	EV-associated extracellular RNA protein-bound miRNA	classical Hodgkin Lymphoma patients had enriched levels of miR24-3p, miR127-3p, miR21-5p, miR155-5p, and let7a-5p, follow-up of EV	[45]
Lymphoma	Chemotherapy	Blood	miRNA PCR	remission in diffuse large B cell Lymphoma was associated with increase of exosomal miR-451a; stable and progressive disease had no significant changes. miRNA revealed stable decrease in miRNA levels.	[46]

CTC, circulating tumor cells; ddPCR, droplet digital PCR; EVs, extracellular vesicles; ELISA, enzyme-linked immunosorbent assay; FACS, fluorescent activated cell sorting; miRNA, microRNA; MPs, microparticles; PEG, polyethylene glycol; UC, ultracentrifuge; SEC, size exclusion chromatography; taMP, tumor-associated microparticles.

## 4. EVs in Glioblastoma

Glioblastoma (GBM) is the most common tumor of the central nervous system (CNS) accounting for 12–15% of all intracranial tumors. Currently, standard care is surgery followed by chemotherapy and radiotherapy [47,48]. Molecular characterization of GBM for classification and further decisions on the treatment are made from biopsy; thus, EVs could be a good tool for both research and clinical purposes. Skog et al. [34] showed that serum EVs (microvesicles) contain messenger RNA mutant/variants and microRNAs characteristic to gliomas. The tumor-specific EGFRvIII was detected in EVs from glioblastoma patients. Thus, this study was one of the first to show that tumor-derived microvesicles may provide diagnostic information and aid in therapeutic decisions for cancer patients through a blood test [34].

Ricklefs et al. [49] previously showed that circulating EV concentrations in GBM patients are elevated. Tumor-specific EVs were detected in a syngeneic mouse tumor model [49]. Osti et al. [35] (Table 1) analyzed plasma EVs (microvesicles) in preoperative GBM samples. EVs concentration significantly declined after resection of the primary GBM and raised with tumor relapse. It was shown that EV concentration in recurrent GBM was nearly 40% higher than in primary GBM samples at the immediate post-resection assessment. The authors suggested the combination of a significant decrease in circulating EVs after GBM removal and EVs enrichment at relapse to be a direct link between EVs and the presence of a GBM mass. Characterizing the protein cargo of plasma EVs in patients with GBM and healthy controls through a MS-based proteomic analysis revealed a GBM-distinctive signature. These findings indicated that measurement of plasma EV concentrations together with the possibility to characterize their specific cargo can be of assistance in diagnosis, treatment, and follow-up of GBM patients [35].

## 5. EVs in Prostate Cancer

In the urological field of prostate cancer (PCa), the most commonly diagnosed cancer in men, the diagnostic approach of analyzing EVs as possible biomarker has gained in significance. Especially the idea of extracting EVs from urine seems to be a legitimate approach to differentiate between urologic malignancy, acute prostatitis, and benign hyperplasia [50]. In 2019, the food and drug administration (FDA) approved the first exosome-based liquid biopsy test, ExoDx™ Prostate IntelliScore (Exosome Diagnostics, Inc., Waltham, MA, USA), analyzing exosomal RNA for three biomarkers (PCA3, TMPRSS2:ERG, SPDEF) on urine specimen [51]. Nonetheless, the prostate specific antigen (PSA) has undeniable clinical importance and is, therefore, used as a diagnostic and prognostic tool. Monitoring PSA and digital rectal examination (DRE) is the mainstay of surveillance testing in men who have undergone definitive therapy for localized PCa. Transrectal ultrasound (TRUS), computed tomography (CT), and positron emission tomography (PET) have no role as screening tests for recurrence of localized prostate cancer [52].

Malla et al. (Table 1) [37] investigated the use of circulating EVs (exosomes) from patients undergoing radiotherapy to monitor treatment response. Radiotherapy (RT) is one primary therapy option for non-metastatic PCa [53]. Nano tracking analysis (NTA) showed higher vesicle concentration of exosomes in patient samples possibly indicating radiation specific induction. Extraction of five shortlisted miRNA relevant to PCa and radiotherapy (RT) (hsa-let-7a-5p, hsa-miR-141-3p, hsa-miR-145-5p, hsa-miR-21-5p, hsa-miR-99b-5p) was performed via qRT-PCR. Upregulation of hsa-let-7a-5p (fold change 2.24) and hsa-miR-21-5p (fold change 1.77) was interpreted as potentially indicating an induction due to radiation [54].

Another approach of monitoring RT response was carried out by Yu et al. [36] (Table 1). Blood extracted EVs and their miRNA cargo were analyzed by next-generation sequencing before and after RT. The 57 miRNAs were found significantly altered after RT. High expression of miRNAs (miR-493-5p, miR-323a-3p, miR-411-5p, miR-494-3p, miR-379-5p, miR-654-3p, miR-409-3p, miR-543, and miR-200c-3p) before RT indicated better therapeutic outcomes. The absolute change in miR-654-3p and miR-379-5p expression was significantly different in a good response (PSA ≤ 0.2 ng/mL after RT) and a poor response (PSA > 0.2 ng/mL after RT) group and, therefore, associated with beneficial RT response [36].

For advanced PCa, current treatments focus on inhibition of the androgen receptor. Soekmadji et al. [38] (Table 1) investigated EV’s protein cargo in supernatant of prostate cancer cells after incubation with enzalutamide. The 34 vesicular proteins were described altered by the androgen receptor antagonist. Especially the plasma membrane calcium pump, ATP2B1/PMCA ATPase, was identified as a part of the cross-talk of androgen receptor signaling and EV pathways in mediating resistance towards hormone therapy [38].

In conclusion, the above-mentioned studies indicate the potential of EVs as a tool to monitor PCa therapy. At present, an increase in EV concentration and miRNA loading show promising results in predicting therapy efficacy for PCa.

## 6. EVs in Colorectal Cancer

Colorectal cancer (CRC) is a common disease and associated with high mortality worldwide. At present, surgical resection combined with adjuvant or neoadjuvant chemo-radiation are available therapy options [55]. In treatment planning, post-treatment follow-up, and in the assessment of prognosis carcinoembryonic antigen (CEA), blood level measurement has significant value [56]. Nevertheless, there is increasing interest in developing methods to improve monitoring accuracy for post-operative recurrence and for therapy monitoring of non-operative options. Willms et al. [40] (Table 1) could show that curative total tumor R0 resection altered tumor-associated microparticles (taMPs). The group analyzed EpCAM and CD147, as common cancer antigens, on taMPs in 52 CRC patients. An increase of EpCAM+ and CD147+ taMPs in comparison to a healthy control group was seen. MPs correlated with the tumor-volume. At 7 days post-op, Willms et al. saw a significant decrease of MPs [40].

A similar approach published by Julich-Haertel et al. [39] (Table 1) also showed EpCAM and CD147 increased on taMPs extracted from 172 patients with liver cancer (hepatocellular carcinoma, HCC; cholangiocarcinoma, CCA). MPs positive for AnnexinV, EpCAM, and ASGPR1 decreased 7 days after curative R0 tumor resection, suggesting close correlation with tumor presence [39]. Taken together, these results suggest a link between decrease and therapy success, implicating that EpCAM+ MPs can be used to monitor therapy response. Santasusagna et al. [41] (Table 1) evaluated the potential of vesicular miRNA to predict therapeutic success in 50 CRC resected patients. Expression levels of miR-200 family was analyzed in EVs (from peripheric and mesenteric blood) and correlated with overall survival (OS). The group saw an association between low levels of miR-200c and miR-14 and significant longer OS, thus identifying two vesicular miRNAs as predictor for therapy failure and poor prognosis [41]. If primarily not resectable in a curative manner, neoadjuvant chemo-radiation therapy (CRT) can be followed by response assessment to evaluate surgical options.

Kassam et al. [42] (Table 1) explored the potential of imaging combined with analysis of liquid biopsy samples to determine therapy response following neoadjuvant CRT. Including 12 patients at baseline and at 4–6 weeks following completion of radiation therapy, circulating tumor cells (CTCs), cell fragments and microparticles (MP) were analyzed. The relative change in the total numbers CTCs, MPs, and cell fragments together with perfusion CT scan accurately predicted tumor response and thus differentiated between responders or nonresponders to neoadjuvant CRT [42]. Another entity of gastrointestinal malignancy is pancreatic cancer (PaCa). Melo et al. [43] (Table 1) evaluated a cell surface proteoglycan, glypican-1 (GPC1) on EVs from serum of PaCa patients at pre- and post-surgery stages. They showed a significant decrease following surgical resection, implicating that GPC1+ EVs (exosomes) may serve as a noninvasive biomarker and a potential monitoring tool to detect therapy response [43]. Another attempt to investigate EVs in PaCa was performed by Bernard et al. [44] (Table 1). Digital droplet PCR was used to determine Kirsten rat sarcoma viral oncogene homolog (KRAS) mutation in vesicular DNA from 34 patients undergoing neoadjuvant treatment for localized PaCa. An increase in vesicular KRAS mutation after therapy was significantly associated with disease progression and not an option for surgical intervention. In contrary, a reduction correlated with resectability. Summarizing vesicular KRAS mutation could serve as predictor for the possibility of a surgical resection after neoadjuvant therapy in PaCa patients [44]. Several studies provide strong evidence for the potential of EVs in gastrointestinal cancer not only to identify progression but also to monitor different therapies.

## 7. EVs in Nonsolid Cancers

EV-associated miRNAs were investigated in classical Hodgkin lymphoma (cHL) patients, and van Eijndhoven et al. [45] (Table 1) found that EV-associated miR21-5p, miR127-3p, let7a-5p, miR24-3p, and miR155-5p were upregulated in primary and relapsed cHL patients compared to healthy controls. They monitored the EV miRNA levels in patients before treatment, directly after treatment, and during long-term follow-up. The levels of miRNA decreased after treatment and it was observed that miRNA level were higher in relapse patients. This study showed that EV-associated miRNAs can be used to monitor therapy response and relapse monitoring in individual cHL patients [45]. Another study from Cao et al. [46] (Table 1) analyzed EV-associated miR-451a to monitor therapy response in diffuse large B cell lymphoma. It was shown that miR-451a was downregulated in patients with diffuse large B cell Lymphoma (DLBCL) compared to healthy controls. As in the previous study, levels of EV-associated miR-451a were analyzed in patients before, during, and after treatment. In the patients who entered complete or partial remission, it was observed that the levels of miR-451a gradually increased. In contrast, patients without remission had no significant change in the levels of miR-451a. Circulating exosomal miR-451a may be a potential indicator for therapy response monitoring in DLBCL [46].

## 8. Conclusions

EVs in different body fluids reflect the overall condition of the patient. Thus, EVs offer a powerful tool for screening, diagnosis, and therapy monitoring of malignancy. The presence of cancer-derived components that are being protected from degradation makes EVs a superior source for liquid biopsy approaches. However, in the field of EV research, the unified terminology, although being addressed by the international society for extracellular vesicles (ISEV) community, is still a problem in addition to nonstandardized protocols for EV isolation and characterization. Therefore, interstudy variations hinder reproducibility and occur due to individual analysis approaches of laboratories working with EVs and their comparison can be critical. Therefore, tools to align EV research are being established. An EV-TRACK database was developed with the goal to facilitate standardization of EV research through increased systematic reporting on EV biology and methodology [57]. The International Society for Extracellular Vesicles (ISEV, http://www.isev.org) proposed Minimal Information for Studies of Extracellular Vesicles (“MISEV”) guidelines [1].

Nevertheless, increasing evidence showed that EVs do carry tumor-specific mutations in nucleic acids and proteins and that EV biomolecular profiles reflect tumor cell changes during therapy in the circulation of patients. If EVs prove to be a reliable source for cancer-derived components, the debate on terminology and standardized isolation techniques cannot exclude its use for liquid biopsy approaches that might have dramatic consequences on how we treat and monitor cancer patients.

## Figures and Tables

**Figure 1 cells-09-00130-f001:**
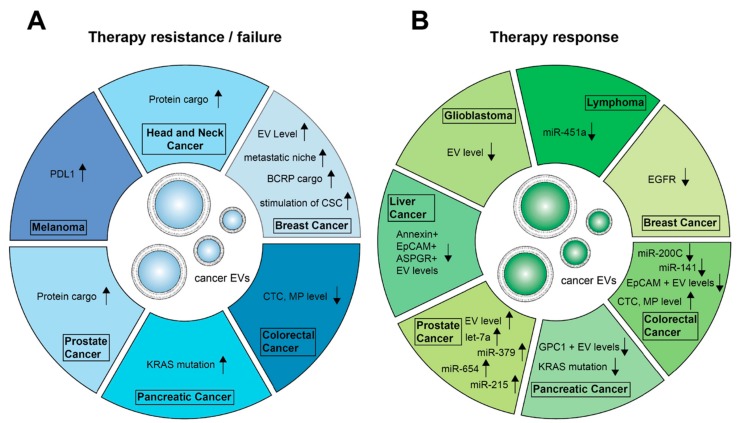
Approaches to monitor therapies by EVs and their cargo. Shown changes were associated with therapy resistance or failure (**A**) in contrast to changes seen if the cancer is responding to therapy (**B**) ↑: increase, upregulation; ↓: decrease, downregulation; EV level↑: Increase in EV concentration; miR: microRNA; CTC: circulation tumor cells; CSC: cancer stem-like cells; MP: microparticles.

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
