# Peer review of "Monitoring Therapy Efficiency in Cancer through Extracellular Vesicles"

_cells, 2020, doi:10.3390/cells9010130_

Round 1

Reviewer 1 Report

Authors have very well discussed the potential role of EVs in predicting the efficacy of chemotherapeutics in different cancers.

it would be nice to present the hypothetical model depicting (diagram) the role of EVs in  predicting the chemotherapeutic efficacy.

Author Response

Authors have very well discussed the potential role of EVs in predicting the efficacy of chemotherapeutics in different cancers.

it would be nice to present the hypothetical model depicting (diagram) the role of EVs in  predicting the chemotherapeutic efficacy.

We added Figure 1 in which we summarized the role of EVs in predicting the chemotherapy response.

Reviewer 2 Report

The paper deals with a very popular topic and may merit the researchers interested in the relevant fields. However, the paper is too unorganized to be understood and therefore, need extensive modifications.

#1. The authors described that extracellular vesicles consist of heterogeneous members but used various terminologies interchangeably. The authors should explicitly explain which type of extracellular vesicle is discussed in each paper.

#2. The section on immune checkpoint was placed between the sections on breast cancer and glioblastoma, making the paper more unorganized. The authors should place the section on immune checkpoint before discussing the individual types of cancers.

#3. The authors should insert the reference numbers in the text at appropriate positions, to help readers check the contents of the cited papers.

#4. The authors showed the summary of up to 20 papers in Table 1 by assigning reference numbers independently of the reference numbers in the main text. The authors should unify the reference numbers.

#5. Each section should be divided into several paragraphs to increase the readability.

#6. The paper contains many sentences which cannot be understood easily and therefore, should be extensively edited by a professional editor proficient in scientific English.

Author Response

Comments and Suggestions for Authors

The paper deals with a very popular topic and may merit the researchers interested in the relevant fields. However, the paper is too unorganized to be understood and therefore, need extensive modifications.

#1. The authors described that extracellular vesicles consist of heterogeneous members but used various terminologies interchangeably. The authors should explicitly explain which type of extracellular vesicle is discussed in each paper.

We are sorry for the misunderstanding. We have now added in brackets which subgroup of EVs was used in the studies if the term EVs was not used.

#2. The section on immune checkpoint was placed between the sections on breast cancer and glioblastoma, making the paper more unorganized. The authors should place the section on immune checkpoint before discussing the individual types of cancers.

We moved the section on immune checkpoint is now before breast cancer (p.2).

#3. The authors should insert the reference numbers in the text at appropriate positions, to help readers check the contents of the cited papers.

We inserted the reference numbers at the required places.

#4. The authors showed the summary of up to 20 papers in Table 1 by assigning reference numbers independently of the reference numbers in the main text. The authors should unify the reference numbers.

We have updated the table with the correct reference numbers (p.4).

#5. Each section should be divided into several paragraphs to increase the readability.

We have divided each section into multiple paragraphs to improve readability

#6. The paper contains many sentences which cannot be understood easily and therefore, should be extensively edited by a professional editor proficient in scientific English.

We have corrected and rephrased our manuscript extensively to improve readability.

Reviewer 3 Report

The review, written by Stevic and her colleagues, explains the distinguished position of cell derived small particles, so called extracellular vesicles, among recent research topics, and reveals why EVs can be very efficient tool especially in diagnosis as well as monitoring cancer therapies.

Broad comments 

Indeed, "the analysis of circulating EV cargo is a potential tool to monitor tumor cell changes during anti-tumor therapy", liquid biopsy is a great, effective, non invasive possibility for getting reliable information from the effectiveness of the therapy, however, the promised discussion is missing.
The reader gets lots of recent and promising results on the research fields of selected cancer types - the Table is very useful -, but the closing sentences tell the same : "Further studies are needed."

It is a general and very frequently used statement in research papers, however, the usefulness of a review is based not only on the compiled information and papers, but on the way of the discussion of their pros and cons. 
In the field of EV research the unified terminology is still a problem and due to the different methods used by the different laboratories, the comparison of the results can be risky.
This is fact, unfortunately, but in some term we can separate the basic research and clinical usefulness of a test, and, luckily, if a test is reliable for monitoring the level and types of given circulating EVs, the debate on terminology cannot exclude its use for for diagnostic purposes.

Taken together, it is  a great topic and the reviewers should take the risk sometimes if they want to introduce the lots of advantages and pitfalls of the use of EVS in cancer diagnosis and therapy

Some specific comments :

line 9: "vesicles made of a phospholipid bilayer and are believed to be secreted by all cell types"    - use of another word can be more proper 

line 10: " RNA, non-coding RNAs"  - while only these are mentioned?

line 20-21: "EVs are a heterogeneous population of lipid bilayer membrane vesicles that are shed by almost all cell types" -  exosomes belong to EVs and they are released 

line 31: "The Liquid biopsy" - not capital "L"

line 42: "investigation of RNA species and their potential as liquid biopsy is evident and has been achieved by numerous studies [12]. " This reference is not enough . ..  

line 59-60: "Nevertheless clinical and pathological assessment after NACT differ notably." Please, clarify this statement briefly.

line 85 : "The authors suspected a mechanism by which cancer cells communicate  with following self-adaption to survive cytotoxic treatment."

A couple of words about the mechanism would be useful - and likely another word instead of "suspected"

line 97: ". Flotillin-2 is an EV marker," - a short description of the role of flotillin-2 could help to understand the significance of this finding

line 141: "Glioblastoma (GBM) is the most common tumor of the central nervous system (CNS) accounting  for 12%–15% of all intracranial tumors."

It is OK, however, as one of the very first article in this area, the paper of Skog et al Nature Cell Biology10 (12): 1470–6. doi:10.1038/ncb1800 -could have been mentioned

Author Response

Comments and Suggestions for Authors

The review, written by Stevic and her colleagues, explains the distinguished position of cell derived small particles, so called extracellular vesicles, among recent research topics, and reveals why EVs can be very efficient tool especially in diagnosis as well as monitoring cancer therapies.

Broad comments 

Indeed, "the analysis of circulating EV cargo is a potential tool to monitor tumor cell changes during anti-tumor therapy", liquid biopsy is a great, effective, non invasive possibility for getting reliable information from the effectiveness of the therapy, however, the promised discussion is missing.
The reader gets lots of recent and promising results on the research fields of selected cancer types - the Table is very useful -, but the closing sentences tell the same : "Further studies are needed." It is a general and very frequently used statement in research papers, however, the usefulness of a review is based not only on the compiled information and papers, but on the way of the discussion of their pros and cons. 
In the field of EV research the unified terminology is still a problem and due to the different methods used by the different laboratories, the comparison of the results can be risky. 
This is fact, unfortunately, but in some term we can separate the basic research and clinical usefulness of a test, and, luckily, if a test is reliable for monitoring the level and types of given circulating EVs, the debate on terminology cannot exclude its use for for diagnostic purposes.

Taken together, it is a great topic and the reviewers should take the risk sometimes if they want to introduce the lots of advantages and pitfalls of the use of EVS in cancer diagnosis and therapy

We thank the reviewer for the comment and completely agree with the above mentioned topic. However, it was our goal to present the data of recent studies on treatment monitoring in cancer as they are and unbiased. The EV research field is moving at such a high speed that it is yet not possible to combine all available data as the methods vary extensively. Therefore the ISEV community is currently proposing research standards to improve reproducibility and also comparability. However, we extensively rephrased our conclusion (p.10)

Some specific comments :

line 9: "vesicles made of a phospholipid bilayer and are believed to be secreted by all cell types"    - use of another word can be more proper 

We corrected our sentence: “Extracellular vesicles (EVs) are a heterogeneous group of membrane-enclosed vesicles made of aphospholipid bilayer and are secreted by all cell types”(p.1)

line 10: " RNA, non-coding RNAs"  - while only these are mentioned?

We apologies for the confusion. We corrected it to …RNA species… (p.1)

line 20-21: "EVs are a heterogeneous population of lipid bilayer membrane vesicles that are shed by almost all cell types" -  exosomes belong to EVs and they are released 

We apologies for not using the proper terminology. We corrected our statement (p.1)

line 31: "The Liquid biopsy" - not capital "L"

We rephrased our sentence (p.1.)

line 42: "investigation of RNA species and their potential as liquid biopsy is evident and has been achieved by numerous studies [12]. " This reference is not enough . ..  

We included additional references. (p.1)

line 59-60: "Nevertheless clinical and pathological assessment after NACT differ notably." Please, clarify this statement briefly.

We clarified this statement as followed:” The correlation between clinical assessment and pathological analysis after NACT is modest at best. Several studies showed a low sensitivity to predict pCR by clinical examination and imaging [23]. Due to the lack of concordance a need for reliable biomarkers that allow non-invasive therapy monitoring and stratification of patients becomes evident.” (p.2 line 93)

line 85 : "The authors suspected a mechanism by which cancer cells communicate  with following self-adaption to survive cytotoxic treatment."

A couple of words about the mechanism would be useful - and likely another word instead of "suspected"

We have explained the mechanism with following sentences “The authors show a mechanism through which chemotherapy-treated breast cancer cells, by secreting certain EV miRNAs, communicate with and reprogram nearby cancer cells to induce a CSC phenotype.Since there are well known associations between CSCs and tumor refractoriness, this mechanism may serve as a means for cancer's self-adaptation to survive the therapy, and may contribute to chemotherapy-induced tumor progression and metastasis.” (p.3 line 126)

line 97: ". Flotillin-2 is an EV marker," - a short description of the role of flotillin-2 could help to understand the significance of this finding

We added the following sentences “Flotillin-2, has been shown to be involved in various cellular processes such as cell adhesion, signal transduction through receptor tyrosine kinases as well as in cellular trafficking pathways. p. 3, line 144

line 141: "Glioblastoma (GBM) is the most common tumor of the central nervous system (CNS) accounting  for 12%–15% of all intracranial tumors."

It is OK, however, as one of the very first article in this area, the paper of Skog et al Nature Cell Biology10 (12): 1470–6. doi:10.1038/ncb1800 -could have been mentioned

We agree that Skog et al revolutionized GBM EVs and we included their paper as a reference.

Round 2

Reviewer 2 Report

The authors modified the manuscript fully in response to the comments.